# A Dual Algorithm for Olfactory Computation in the Locust Brain

**Sina Tootoonian**            **Máté Lengyel**
st582@eng.cam.ac.uk   m.lengyel@eng.cam.ac.uk

Computational & Biological Learning Laboratory
Department of Engineering, University of Cambridge
Trumpington Street, Cambridge CB2 1PZ, United Kingdom

## Abstract

We study the early locust olfactory system in an attempt to explain its well-characterized structure and dynamics. We first propose its computational function as recovery of high-dimensional sparse olfactory signals from a small number of measurements. Detailed experimental knowledge about this system rules out standard algorithmic solutions to this problem. Instead, we show that solving a dual formulation of the corresponding optimisation problem yields structure and dynamics in good agreement with biological data. Further biological constraints lead us to a reduced form of this dual formulation in which the system uses independent component analysis to continuously adapt to its olfactory environment to allow accurate sparse recovery. Our work demonstrates the challenges and rewards of attempting detailed understanding of experimentally well-characterized systems.

## 1 Introduction

Olfaction is perhaps the most widespread sensory modality in the animal kingdom, often crucial for basic survival behaviours such as foraging, navigation, kin recognition, and mating. Remarkably, the neural architecture of olfactory systems across phyla is largely conserved [1]. Such convergent evolution suggests that what we learn studying the problem in small model systems will generalize to larger ones. Here we study the olfactory system of the locust *Schistocerca americana*. While we focus on this system because it is experimentally well-characterized (Section 2), we expect our results to extend to other olfactory systems with similar architectures. We begin by observing that although most odors are mixtures of hundreds of molecular species, with typically only a few of these dominating in concentration – i.e. odors are *sparse* in the space of molecular concentrations (Fig. 1A). We introduce a simple generative model of odors and their effects on odorant receptors that reflects this sparsity (Section 3). Inspired by recent experimental findings [2], we then propose that the function of the early olfactory system is maximum *a posteriori* (MAP) inference of these concentration vectors from receptor inputs (Section 4). This is basically a sparse signal recovery problem, but the wealth of biological evidence available about the system rules out standard solutions. We are then led by these constraints to propose a novel solution to this problem in term of its dual formulation (Section 5), and further to a reduced form of this solution (Section 6) in which the circuitry uses ICA to continuously adapt itself to the local olfactory environment (Section 7). We close by discussing predictions of our theory that are amenable to testing in future experiments, and future extensions of the model to deal with readout and learning simultaneously, and to provide robustness against noise corrupting sensory signals (Section 8).

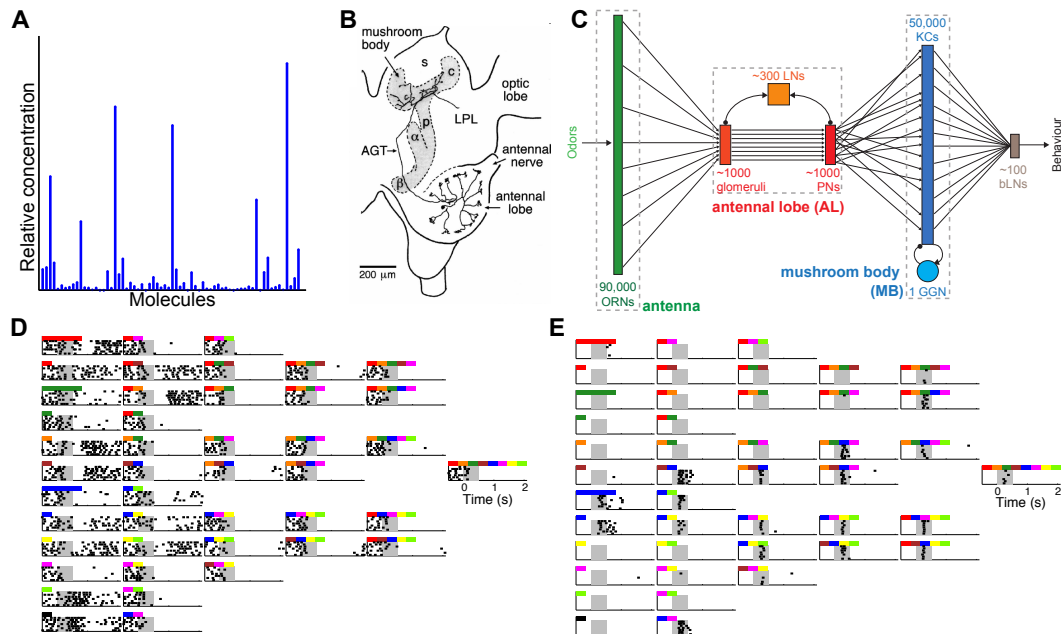

Figure 1: Odors and the olfactory circuit. **(A)** Relative concentrations of $\sim 70$ molecules in the odor of the Festival strawberry cultivar, demonstrating sparseness of odor vectors. **(B,C)** Diagram and schematic of the locust olfactory circuit. Inputs from 90,000 ORNs converge onto $\sim 1000$ glomeruli, are processed by the $\sim 1000$ cells (projection neurons, PN, and local internuerons, LNs) of the antennal lobe, and read out in a feedforward manner by the 50,000 Kenyon cells (KC) of the mushroom body, whose activity ultimately is read out to produce behavior. **(D,E)** Odor response of a PN (D) and a KC (E) to 7 trials of 44 mixtures of 8 monomolecular components (colors) demonstrating cell- and odor-specific responses. The odor presentation window is in gray. PN responses are dense and temporally patterned. KC responses are sparse and are often sensitive to single molecules in a mixture. Panel A is reproduced from [8], B from [6], and D-E from the dataset in [2].

## 2 Biological background

A schematic of the locust olfactory system is shown in Figure 1B-C. Axons from $\sim 90,000$ olfactory receptor neurons (ORNs) each thought to express one type of olfactory receptor (OR) converge onto approximately 1000 spherical neuropilar structures called 'glomeruli', presumably by the '1-OR-to-1-glomerulus' rule observed in flies and mice. The functional role of this convergence is thought to be noise reduction through averaging.

The glomeruli are sampled by the approximately 800 excitatory projection neurons (PNs) and 300 inhibitory local interneurons (LNs) of the antennal lobe (AL). LNs are densely connected to other LNs and to the PNs; PNs are connected to each-other only indirectly via their dense connections to LNs [3]. In response to odors, the AL exhibits 20 Hz local field potential oscillations and odor- and cell-specific activity patterns in its PNs and LNs (Fig. 1D). The PNs form the only output of the AL and project densely [4] to the 50,000 Kenyon cells (KCs) of the mushroom body (MB). The KCs decode the PNs in a memoryless fashion every oscillation cycle, converting the dense and promiscuous PN odor code into a very sparse and selective KC code [5], often sensitive to a single component in a complex odor mixture [2] (Fig. 1E). KCs make axo-axonal connections with neighbouring KCs [6] but otherwise only communicate with one-another indirectly via global inhibition mediated by the giant GABA-ergic neuron [7]. Thus, while the AL has rich recurrency, there is no feedback from the KCs back to the AL: the PN to KC circuit is strictly feedforward. As we shall see below, this presents a fundamental challenge to theories of AL-MB computation.

## 3  Generative model

Natural odors are mixtures of hundreds of different types of molecules at various concentrations (e.g. [8]), and can be represented as points in $\mathbb{R}_+^N$, where each dimension represents the concentration of one of the $N$ molecular species in 'odor space'. Often a few of these will be at a much higher concentration than the others, i.e. natural odors are sparse. Because the AL responds similarly across concentrations [9] , we will ignore concentration in our odor model and consider odors as binary vectors $\mathbf{x} \in \{0,1\}^N$. We will also assume that molecules appear in odor vectors independently of one-another with probability $k/N$, where $k$ is the average complexity of odors (# of molecules/odor, equivalently the Hamming weight of $\mathbf{x}$) in odor space.

We assume a linear noise-free observation model $\mathbf{y} = \mathbf{A}\mathbf{x}$ for the $M$-dimensional glomerular activity vector (we discuss observation noise in Section 7). $\mathbf{A}$ is an $M \times N$ affinity matrix representing the response of each of the $M$ glomeruli to each of the $N$ molecular odor components and has elements drawn iid. from a zero-mean Gaussian with variance $1/M$. Our generative model for odors and observations is summarized as

$$\mathbf{x} = \{x_1, \ldots, x_N\},\ x_i \sim \text{Bernoulli}(k/N), \quad \mathbf{y} = \mathbf{A}\mathbf{x}, \quad A_{ij} \sim \mathcal{N}(0, M^{-1}) \tag{1}$$

## 4  Basic MAP inference

Inspired by the sensitivity of KCs to monomolecular odors [2], we propose that the locust olfactory system acts as a spectrum analyzer which uses MAP inference to recover the sparse $N$-dimensional odor vector $\mathbf{x}$ responsible for the dense $M$-dimensional glomerular observations $\mathbf{y}$, with $M \ll N$ e.g. $\mathcal{O}(1000)$ vs. $\mathcal{O}(10000)$ in the locust. Thus, the computational problem is akin to one in compressed sensing [10], which we will exploit in Section 5. We posit that each KC encodes the presence of a single molecular species in the odor, so that the overall KC activity vector represents the system's estimate of the odor that produced the observations $\mathbf{y}$.

To perform MAP inference on binary $\mathbf{x}$ from $\mathbf{y}$ given $\mathbf{A}$, a standard approach is to relax $\mathbf{x}$ to the positive orthant $\mathbb{R}_+^N$ [11], smoothen the observation model with isotropic Gaussian noise of variance $\sigma^2$ and perform gradient descent on the log posterior

$$\log p(\mathbf{x}|\mathbf{y}, \mathbf{A}, k) = C - \beta\|\mathbf{x}\|_1 - \frac{1}{2\sigma^2}\|\mathbf{y} - \mathbf{A}\mathbf{x}\|_2^2 \tag{2}$$

where $\beta = \log((1-q)/q)$, $q = k/N$, $\|\mathbf{x}\|_1 = \sum_{i=1}^M \mathbf{x}_i$ for $\mathbf{x} \succeq 0$, and $C$ is a constant. The gradient of the posterior determines the $\mathbf{x}$ dynamics:

$$\dot{\mathbf{x}} \propto \nabla_{\mathbf{x}} \log p = -\beta\,\text{sgn}(\mathbf{x}) + \frac{1}{2\sigma^2}\mathbf{A}^T(\mathbf{y} - \mathbf{A}\mathbf{x}) \tag{3}$$

Given our assumed 1-to-1 mapping of KCs to (decoded) elements of $\mathbf{x}$, these dynamics fundamentally violate the known biology for two reasons. First, they stipulate KC dynamics where there are none. Second, they require all-to-all connectivity of KCs via $\mathbf{A}^T\mathbf{A}$ where none exist. In reality, the dynamics in the circuit occur in the lower ($\sim M$) dimensional measurement space of the antennal lobe, and hence we need a way of solving the inference problem there rather than directly in the high ($N$) dimensional space of KC activites.

## 5  Low dimensional dynamics from duality

To compute the MAP solution using lower-dimensional dynamics, we consider the following compressed sensing (CS) problem:

$$\text{minimize}\ \|\mathbf{x}\|_1, \quad \text{subject to } \|\mathbf{y} - \mathbf{A}\mathbf{x}\|_2^2 = 0 \tag{4}$$

whose Lagrangian has the form

$$L(\mathbf{x}, \lambda) = \|\mathbf{x}\|_1 + \lambda\|\mathbf{y} - \mathbf{A}\mathbf{x}\|_2^2 \tag{5}$$

where $\lambda$ is a scalar Lagrange multiplier. This is exactly the equation for our (negative) log posterior (Eq. 2) with the constants absorbed by $\lambda$. We will assume that because $\mathbf{x}$ is binary, the two systems will have the same solution, and will henceforth work with the CS problem.

To derive low dimensional dynamics, we first reformulate the constraint and solve

$$\text{minimize } \|\mathbf{x}\|_1, \quad \text{subject to } \mathbf{y} = \mathbf{Ax} \tag{6}$$

with Lagrangian

$$L(\mathbf{x}, \boldsymbol{\lambda}) = \|\mathbf{x}\|_1 + \boldsymbol{\lambda}^T(\mathbf{y} - \mathbf{Ax}) \tag{7}$$

where now $\boldsymbol{\lambda}$ is a vector of Lagrange multipliers. Note that we are still solving an $N$-dimensional minimization problem with $M \ll N$ constraints, while we need $M$-dimensional dynamics. Therefore, we consider the dual optimization problem of maximizing $g(\boldsymbol{\lambda})$ where $g(\boldsymbol{\lambda}) = \inf_{\mathbf{x}} L(\mathbf{x}, \boldsymbol{\lambda})$ is the *dual Lagrangian* of the problem. If strong duality holds, the primal and dual objectives have the same value at the solution, and the primal solution can be found by minimizing the Lagrangian at the optimal value of $\boldsymbol{\lambda}$ [11]. Were $\mathbf{x} \in \mathbb{R}^N$, strong duality would hold for our problem by Slater's sufficiency condition [11]. The binary nature of $\mathbf{x}$ robs our problem of the convexity required for this sufficiency condition to be applicable. Nevertheless we proceed assuming strong duality holds.

The dual Lagrangian has a closed-form expression for our problem. To see this, let $\mathbf{b} = \mathbf{A}^T\boldsymbol{\lambda}$. Then, exploiting the form of the 1-norm and $\mathbf{x}$ being binary, we obtain the following:

$$g(\boldsymbol{\lambda}) - \boldsymbol{\lambda}^T\mathbf{y} = \inf_{\mathbf{x}} \|\mathbf{x}\|_1 - \mathbf{b}^T\mathbf{x} = \inf_{\mathbf{x}} \sum_{i=1}^{M}(|\mathbf{x}_i| - \mathbf{b}_i\mathbf{x}_i) = \sum_{i=1}^{M}\inf_{\mathbf{x}_i}(|\mathbf{x}_i| - \mathbf{b}_i\mathbf{x}_i) = -\sum_{i=1}^{M}[\mathbf{b}_i - 1]_+ \tag{8}$$

or, in vector form, $g(\boldsymbol{\lambda}) = \boldsymbol{\lambda}^T\mathbf{y} - \mathbf{1}^T[\mathbf{b} - 1]_+$, where $[\cdot]_+$ is the positive rectifying function. Maximizing $g(\boldsymbol{\lambda})$ by gradient descent yields $M$ dimensional dynamics in $\boldsymbol{\lambda}$:

$$\dot{\boldsymbol{\lambda}} \propto \nabla_{\boldsymbol{\lambda}} g = \mathbf{y} - \mathbf{A}\, \theta(\mathbf{A}^T\boldsymbol{\lambda} - \mathbf{1}) \tag{9}$$

where $\theta(\cdot)$ is the Heaviside function. The solution to the CS problem – the odor vector that produced the measurements $\mathbf{y}$ – is then read out at the convergence of these dynamics to $\boldsymbol{\lambda}_\star$ as

$$\mathbf{x}_\star = \text{argmin}_{\mathbf{x}} L(\mathbf{x}, \boldsymbol{\lambda}_\star) = \theta(\mathbf{A}^T\boldsymbol{\lambda}_\star - \mathbf{1}) \tag{10}$$

A natural mapping of equations 9 and 10 to antennal lobe dynamics is for the output of the $M$ glomeruli to represent $\mathbf{y}$, the PNs to represent $\boldsymbol{\lambda}$, and the KCs to represent (the output of) $\theta$, and hence eventually $\mathbf{x}_\star$. Note that this would still require the connectivity between PNs and KCs to be negative reciprocal (and determined by the affinity matrix $\mathbf{A}$). We term the circuit under this mapping the *full dual* circuit (Fig. 2B). These dynamics allow neuronal firing rates to be both positive and negative, hence they can be implemented in real neurons as e.g. deviations relative to a baseline rate [12], which is subtracted out at readout.

We measured the performance of a *full dual* network of $M = 100$ PNs in recovering binary odor vectors containing an average of $k = 1$ to 10 components out of a possible $N = 1000$. The results in Figure 2E (blue) show that the dynamics exhibit perfect recovery.[1] For comparison, we have included the performance of the purely feedforward circuit (Fig. 2A), in which the glomerular vector $\mathbf{y}$ is merely scaled by the $k$-specific amount that yields minimum error before being read out by the KCs (Fig. 2E, black). In principle, no recurrent circuit should perform worse than this feedfoward network, otherwise we have added substantial (energetic and time) costs without computational benefits.

## 6   The reduced dual circuit

The *full dual* antennal lobe circuit described by Equations 9 and 10 is in better agreement with the known biology of the locust olfactory system than 2 for a number of reasons:

1. Dynamics are in the lower dimensional space of the antennal lobe PNs ($\boldsymbol{\lambda}$) rather than the mushroom body KCs ($\mathbf{x}$).

2. Each PN $\boldsymbol{\lambda}_i$ receives private glomerular input $\mathbf{y}_i$

3. There are no direct connections between PNs; their only interaction with other PNs is indirect via inhibition provided by $\theta$.

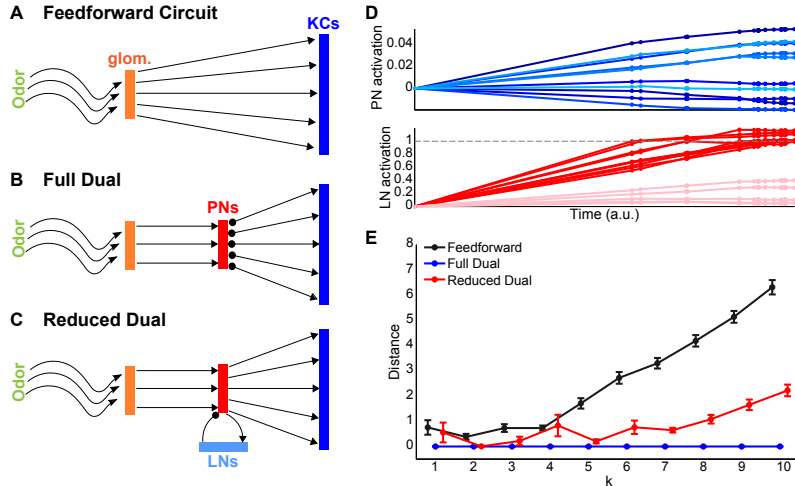

Figure 2: Performance of the feedforward and the dual circuits. **(A-C)** Circuit schematics. Arrows (circles) indicate excitatory (inhibitory) connections. **(D)** Example PN and LN odor-evoked dynamics for the reduced dual circuit. Top: PNs receive cell-specific excitation or inhibition whose strength is changed as different LNs are activated, yielding cell-specific temporal patterning. Bottom: The LNs whose corresponding KCs encode the odor (red) are strongly excited and eventually breach the threshold (dashed line), causing changes to the dynamics (time points marked with dots). The excitation of the other LNs (pink) remains subthreshold. **(E)** Hamming distance between recovered and true odor vector as a function of odor density $k$. The dual circuits generally outperform the feedforward system over the entire range tested. Points are means, bars are s.e.m., computed for 200 trials (feedforward) and all trials from 200 attempts in which the steady-state solution was found (dual circuits, greater than 90%).

4. The KCs serve merely as a readout stage and are not interconnected.[2]

However, there is also a crucial *dis*agreement of the *full dual* dynamics with biology: the requirement for feedback from the KCs to the PNs. The mapping of $\boldsymbol{\lambda}$ to PNs and $\theta$ to the KCs in Equation 9 implies negative reciprocal connectivity of PNs and KCs, i.e. a feedforward connection of $\mathbf{A}_{ij}$ from PN $i$ to KC $j$, and a feedback connection of $-\mathbf{A}_{ij}$ from KC $j$ to PN $i$. This latter connection from KCs to PNs violates biological fact – no such direct and specific connectivity from KCs to PNs exists in the locust system, and even if it did, it would most likely be excitatory rather than inhibitory, as KCs are excitatory.

Although KCs are not inhibitory, antennal lobe LNs *are* and connect densely to the PNs. Hence they could provide the feedback required to guide PN dynamics. Unfortunately, the number of LNs is on the order of that of the PNs, i.e. much fewer than the number of the KCs, making it *a priori* unlikely that they could replace the KCs in providing the detailed pattern of feedback that the PNs require under the *full dual* dynamics.

To circumvent this problem, we make two assumptions about the odor environment. The first is that any given environment contains a small fraction of the set of all possible molecules in odor space. This implies the potential activation of only a small number of KCs, whose feedback patterns (columns of $\mathbf{A}$) could then be provided by the LNs. The second assumption is that the environment changes sufficiently slowly that the animal has time to learn it, i.e. that the LNs can update their feedback patterns to match the change in required KC activations.

This yields the *reduced dual* circuit, in which the reciprocal interaction of the PNs with the KCs via the matrix $\mathbf{A}$ is replaced with interaction with the $M$ LNs via the square matrix $\mathbf{B}$. The activity of the LNs represents the activity of the KCs encoding the molecules in the current odor environment,

and the columns of $\mathbf{B}$ are the corresponding columns of the full $\mathbf{A}$ matrix:

$$\dot{\boldsymbol{\lambda}} \propto \mathbf{y} - \mathbf{B}\,\theta(\mathbf{B}^T\boldsymbol{\lambda} - \mathbf{1}), \quad \mathbf{x} = \theta(\mathbf{A}^T\boldsymbol{\lambda} - 1) \tag{11}$$

Note that instantaneous readout of the PNs is still performed by the KCs as in the *full dual*. The performance of the *reduced dual* is shown in red in Figure 2E, demonstrating better performance than the feedforward circuit, though not the perfect recovery of the *full dual*. This is because the solution sets of the two equations are not the same: Suppose that $\mathbf{B} = \mathbf{A}_{:,1:M}$, and that $\mathbf{y} = \sum_{i=1}^{k} \mathbf{A}_{:,i}$. The corresponding solution set for *reduced dual* is $\Lambda_1(\mathbf{y}) = \{\boldsymbol{\lambda} : (\mathbf{B}_{:,1:k})^T\boldsymbol{\lambda} > 1 \wedge (\mathbf{B}_{:,k+1:M})^T\boldsymbol{\lambda} < 1\}$, equivalently $\Lambda_1(\mathbf{y}) = \{\boldsymbol{\lambda} : (\mathbf{A}_{:,1:k})^T\boldsymbol{\lambda} > 1 \wedge (\mathbf{A}_{:,k+1:M})^T\boldsymbol{\lambda} < 1\}$. On the other hand, the solution set for the *full dual* is $\Lambda_0(\mathbf{y}) = \{\boldsymbol{\lambda} : (\mathbf{A}_{:,1:k})^T\boldsymbol{\lambda} > 1 \wedge (\mathbf{A}_{:,k+1:M})^T\boldsymbol{\lambda} < 1 \wedge (\mathbf{A}_{:,M+1:N})^T\boldsymbol{\lambda} < 1\}$. Note the additional requirement that the projection of $\boldsymbol{\lambda}$ onto columns $M + 1$ to $N$ of $\mathbf{A}$ must also be less than 1. Hence any solution to the *full dual* is a solution to the *reduced dual*, but not necessarily vise-versa: $\Lambda_0(\mathbf{y}) \subseteq \Lambda_1(\mathbf{y})$. Since only the former are solutions to the full problem, not all solutions to the *reduced dual* will solve it, leading to the reduced peformance observed. This analysis also implies that increasing (or decreasing) the number of columns in $\mathbf{B}$, so that it is no longer square, will improve (worsen) the performance of the *reduced dual*, by making its solution-set a smaller (larger) superset of $\Lambda_0(\mathbf{y})$.

# 7 Learning via ICA

Figure 2 demonstrates that the *reduced dual* has reasonable performance when the $\mathbf{B}$ matrix is correct, i.e. it contains the columns of $\mathbf{A}$ for the KCs that would be active in the current odor environment. How would this matrix be learned before birth, when presumably little is known about the local environment, or as the animal moves from one odor environment to another?

Recall that, according to our generative model (Section 2) and the additional assumptions made for deriving the *reduced dual* circuit (Section 6), molecules appear independently at random in odors of a given odor environment and the mapping from odors $\mathbf{x}$ to glomerular responses $\mathbf{y}$ is linear in $\mathbf{x}$ via the square mixing matrix $\mathbf{B}$. Hence, our problem of learning $\mathbf{B}$ is precisely that of ICA (or more precisely, sparse coding, as the observation noise variance is assumed to be $\sigma^2 > 0$ for inference), with binary latent variables $\mathbf{x}$. We solve this problem using MAP inference via EM with a mean-field variational approximation $q(\mathbf{x})$ to the posterior $p(\mathbf{x}|\mathbf{y}, \mathbf{B})$ [13], where $q(\mathbf{x}) \triangleq \prod_{i=1}^{M} \text{Bernoulli}(\mathbf{x}_i; \mathbf{q}_i) = \prod_{i=1}^{M} \mathbf{q}_i^{\mathbf{x}_i}(1 - \mathbf{q}_i)^{1-\mathbf{x}_i}$. The E-step, after observing that for binary $x$, $x^2 = x$, is $\Delta\mathbf{q} \propto -\boldsymbol{\gamma} - \log\frac{\mathbf{q}}{1-\mathbf{q}} + \frac{1}{\sigma^2}\mathbf{B}^T\mathbf{y} - \frac{1}{\sigma^2}\mathbf{C}\mathbf{q}$, with $\boldsymbol{\gamma} = \beta\mathbf{1} + \frac{1}{2\sigma^2}\mathbf{c}$, $\beta = \log((1 - q_0)/q_0)$, $q_0 = k/M$, the vector $\mathbf{c} = \text{diag}(\mathbf{B}^T\mathbf{B})$, and $\mathbf{C} = \mathbf{B}^T\mathbf{B} - \text{diag}(\mathbf{c})$, i.e. $\mathbf{C}$ is $\mathbf{B}^T\mathbf{B}$ with the diagonal elements set to zero. To yield more plausible neural dynamics, we change variables to $\mathbf{v} = \log(\mathbf{q}/(1 - \mathbf{q}))$. By the chain rule $\dot{\mathbf{v}} = \text{diag}(\partial\mathbf{v}_i/\partial\mathbf{q}_i)\dot{\mathbf{q}}$. As $\mathbf{v}_i$ is monotonically increasing in $\mathbf{q}_i$, and so the corresponding partial derivatives are all positive, and the resulting diagonal matrix is positive definite, we can ignore it in performing gradient descent and still minimize the same objective. Hence we have

$$\Delta\mathbf{v} \propto -\boldsymbol{\gamma} - \mathbf{v} + \frac{1}{\sigma^2}\mathbf{B}^T\mathbf{y} - \frac{1}{\sigma^2}\mathbf{C}\mathbf{q}(\mathbf{v}), \quad \mathbf{q}(\mathbf{v}) = \frac{1}{1 + \exp(-\mathbf{v})}, \tag{12}$$

with the obvious mapping of $\mathbf{v}$ to LN membrane potentials, and $\mathbf{q}$ as the sigmoidal output function representing graded voltage-dependent transmitter release observed in locust LNs.

The M-step update is made by changing $\mathbf{B}$ to increase $\log p(\mathbf{B}) + \mathbb{E}_q \log p(\mathbf{x}, \mathbf{y}|\mathbf{B})$, yielding

$$\Delta\mathbf{B} \propto -\frac{1}{M}\mathbf{B} + \frac{1}{\sigma^2}(\mathbf{r}\mathbf{q}^T + \mathbf{B}\,\text{diag}(\mathbf{q}(1 - \mathbf{q}))), \quad \mathbf{r} \triangleq \mathbf{y} - \mathbf{B}\mathbf{q}. \tag{13}$$

Note that this update rule takes the form of a local learning rule.

Empirically, we observed convergence within around 10,000 iterations using a fixed step size of $dt \approx 10^{-2}$, and $\sigma \approx 0.2$ for $M$ in the range of 20–100 and $k$ in the range of 1–5. In cases when the algorithm did not converge, lowering $\sigma$ slightly typically solved the problem. The performance of the algorithm is shown in figure 3. Although the $\mathbf{B}$ matrix is learned to high accuracy, it is not learned *exactly*. The resulting algorithmic noise renders the performance of the dual shown in Fig. 2E an upper bound, since there the exact $\mathbf{B}$ matrix was used.

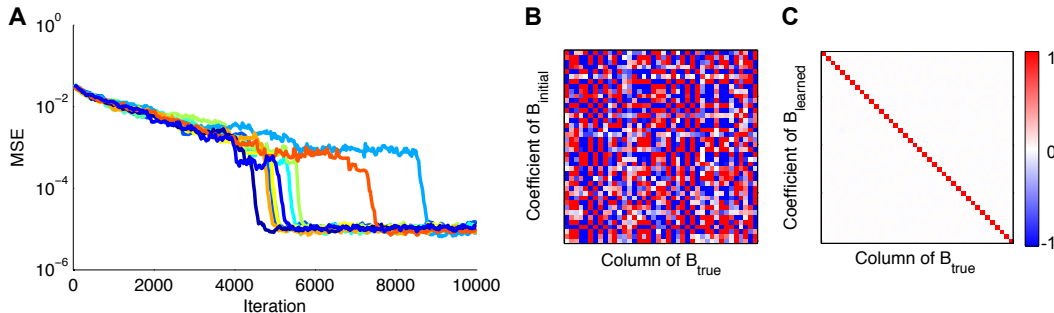

Figure 3: ICA performance for $M = 40$, $k = 1$, $dt = 10^{-2}$. **(A)** Time course of mean squared error between the elements of the estimate $\mathbf{B}$ and their true values for 10 different random seeds. $\sigma = 0.162$ for six of the seeds, $0.15$ for three, and $0.14$ for one. **(B,C)** Projection of the columns of $\mathbf{B}_{true}$ into the basis of the columns of $\mathbf{B}$ before (B) and after learning (C), for one of the random seeds. Plotted values before learning are clipped to the -1–1 range.

## 8 Discussion

### 8.1 Biological evidence and predictions

Our work is consistent with much of the known anatomy of the locust olfactory system, e.g. the lack of connectivity between PNs and dense connectivity between LNs, and between LNs and PNs [3]; direct ORN inputs to LNs (observed in flies [14]; unknown in locust); dense connectivity from PNs to KCs [4]; odor-evoked dynamics in the antennal lobe [2], vs. memoryless readout in the KCs [5]. In addition, we require gradient descent PN dynamics (untested directly, but consistent with PN dynamics reaching fixed-points upon prolonged odor presentation [15]), and short-term plasticity in the antennal lobe for ICA (a direct search for ICA has not been performed, but short-term plasticity is present in trial-to-trial dynamics [16]).

Our model also makes detailed predictions about circuit connectivity. First, it predicts a specific structure for the PN-to-KC connectivity matrix, namely $\mathbf{A}^T$, the transpose of the affinity matrix. This is superficially at odds with recent work in flies suggesting random connectivity between PNs and KCs (detailed connectivity information is not present in the locust). Murthy and colleagues [17] examined a small population of genetically identifiable KCs and found no evidence of response stereotypy across flies, unlike that present at earlier stages in the system. Our model is agnostic to permutations of the output vector as these reassign the mapping between KCs and molecules and affect neither information content nor its format, so our results would be consistent with [17] under animal-specific permutations. Caron and co-workers [18] analysed the structural connectivity of single KCs to glomeruli and found it consistent with random connectivity conditioned on a glomerulus-specific connection probability. This is also consistent with our model, with the observed randomness reflecting that of the affinity matrix itself. Our model would predict (a) the observation of repeated connectivity motifs if enough KCs (across animals) were observed, and that (b) each connectivity motif corresponds to the (binarized) glomerular response vector evoked by a particular molecule. In addition we predict symmetric inhibitory connectivity between LNs ($\mathbf{B}^T \mathbf{B}$), and negative reciprocal connectivity between PNs and LNs ($\mathbf{B}_{ij}$ from PN $i$ to LN $j$ and $-\mathbf{B}_{ij}$ from LN to PN).

### 8.2 Combining learning and readout

We have presented two mechanisms above – the *reduced dual* for readout and and ICA for learning – both of which need to be at play to guarantee high performance. In fact, these two mechanisms must be active simultaneously in the animal. Here we sketch a possible mechanism for combining them. The key is equation 12, which we repeat below, augmented with an additional term from the PNs:

$$\Delta \mathbf{v} \propto -\mathbf{v} + \left[ -\boldsymbol{\gamma} + \frac{1}{\sigma^2} \mathbf{B}^T \mathbf{y} - \frac{1}{\sigma^2} \mathbf{C}\,\mathbf{q}(\mathbf{v}) \right] + \left[ \mathbf{B}^T \boldsymbol{\lambda} - \mathbf{1} \right] = -\mathbf{v} + \mathbf{I}_{\text{learning}} + \mathbf{I}_{\text{readout}}.$$

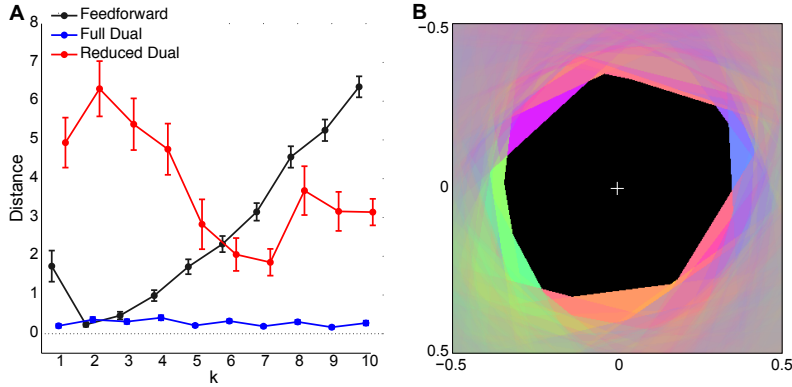

Figure 4: Effects of noise. **(A)** As in Figure 2E but with a small amount of additive noise in the observations. The full dual still outperforms the feedforward circuit which in turn outperforms the reduced dual over nearly half the tested range. **(B)** The feedback surface hinting at noise sensitivity. PN phase space is colored according to activation of each of the KCs and a 2D projection around the origin is shown. The average size of a zone with a uniform color is quite small, suggesting that small perturbations would change the configuration of KCs activated by a PN, and hence the readout performance.

Suppose (a) the two input channels were segregated e.g. on separate dendritic compartments, and such that (b) the readout component was fast but weak, while (c) the learning component was slow but strong, and (d) the $\mathbf{v}$ time constant was faster than both. Early after odor presentation, the main input to the LN would be from the readout circuit, driving the PNs to their fixed point. The input from the learning circuit would eventually catch up and dominate that of the readout circuit, driving the LN dynamics for learning. Importantly, if $\mathbf{B}$ has already been learned, then the output of the LNs, $\mathbf{q}(\mathbf{v})$, would remain essentially unchanged throughout, as both the learning and readout circuits would produce the same (steady-state) activation vector in the LNs. If the matrix is incorrect, then the readout is likely to be incorrect already, and so the important aspect is the learning update which would eventually dominate. This is just one possibility for combining learning and readout. Indeed, even the ICA updates themselves are non-trivial to implement. We leave the details of both to future work.

### 8.3 Noise sensitivity

Although our derivations for serving inference and learning rules assumed observation noise, the data that we provided to the models contained none. Adding a small amount of noise reduces the performance of the dual circuits, particularly that of the reduced dual, as shown in Figure 4A. Though this may partially be attributed to numerical integration issues (Supplementary Material), there is likely a fundamental theoretical cause underlying it. This is hinted at by the plot in figure 4B of a 2D projection in PN space of the overlayed halfspaces defined by the activation of each of the $N$ KCs. In the central void no KC is active and $\boldsymbol{\lambda}$ can change freely along $\dot{\boldsymbol{\lambda}}$. As $\boldsymbol{\lambda}$ crosses into a halfspace, the corresponding KC is activated, changing $\dot{\boldsymbol{\lambda}}$ and the trajectory of $\boldsymbol{\lambda}$. The different colored zones indicate different patterns of KC activation and correspondingly different changes to $\dot{\boldsymbol{\lambda}}$. The small size of these zones suggests that small changes in the trajectory of $\boldsymbol{\lambda}$ caused e.g. by noise could result in very different patterns of KC activation. For the reduced dual, most of these halfspaces are absent for the dynamics since $\mathbf{B}$ has only a small subset of the columns of $\mathbf{A}$, but are present during readout, exacerbating the problem. How the biological system overcomes this apparently fundamental sensitivity is an important question for future work.

**Acknowledgements**    This work was supported by the Wellcome Trust (ST, ML).

## Footnotes

[1] See the the Supplementary Material for considerations when simulating the piecewise linear dynamics of 9.

[2]Although axo-axonal connections between neighbouring KC axons in the mushroom body peduncle are known to exist [6], see also Section 2.

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
