[Supplementary Material]

# A Dual Algorithm for Olfactory Computation in the Locust Brain
# – Supplementary Material –

**Sina Tootoonian**                **Máté Lengyel**
st582@eng.cam.ac.uk    m.lengyel@eng.cam.ac.uk

Computational & Biological Learning Laboratory
Department of Engineering, University of Cambridge
Trumpington Street, Cambridge CB2 1PZ, United Kingdom

**Integration of PN Dynamics**

PN dynamics in both of the dual models are piecewise linear and hence in principle simple to integrate. However, the noise sensitivity of the system (discussed above) means that integration must be done with high accuracy. Since the PN velocity only changes when the KC (or LN) configuration changes, the custom integrator used in this work operates by integrating the dynamics exactly, up to such a change point. The velocity is then updated and the process repeated until either the velocity shrinks below a threshold ($10^{-6}$; length of typical velocity vector: $\mathcal{O}(10)$), a maximum number of change points (500) have been observed, or a maximum integration time ($\mathcal{O}(0.1$ sec)) has been exceeded. To avoid oscillations around change points, the integration is performed to a distance slightly ($10^{-8}$) beyond it, and additionally KCs (or LNs) have a refactory period ($10^{-8}$ sec) to avoid spurious state changes. The integrator is very fast and works reasonably well in that it can typically find the steady state solution in the noise-free case, while in the noisy case (where a steady state solution does not exist), the system is typically able to integrate up to the maximum allotted time without an excessive number of state changes. It would be important for future work to develop better criteria for measuring performance, especially in the noisy case, and more generally to determine whether a more accurate integration method exists for this type of problem.