[Reviews · NeurIPS 2014]

Submitted by Assigned_Reviewer_4

Summary:
This paper attempts to link sparse optimization methodology to the anatomical structure of locust's early
olfactory system. The work is motivated by the observation that odorant molecules are sparsely represented
by the population of Kenyon cells. The authors first mathematically formulate the olfactory system as a MAP
decoder, and give the standard solution to the problem without considering biological constraints. Next, to
make the solution more biologically plausible, the authors reformulate the olfactory system model as a decoder
of a compressive sensing problem, and provide two standard solutions to the dual problem. Then, the authors
argue that each of the components in the solution can be mapped/interpreted to/as a unit of the biological
structure in the olfactory system. However, these maps are described without a strong justification
and there are conceptual problems in linking the math with the biology.

Further clarification on the following points would help readers to gain better insights:

- While LNs and KCs are modeled using a Heaviside function, modeling that is widely accepted in the literature,
the PNs output is modeled as the Lagrange multiplier. Such modeling is not intuitive, and needs to be justified.
In particular, under such modeling, the output of PNs is computed by the gradient decent method in Eq. (10) for
the full dual algorithm, and in Eq. (16) for the reduced dual, which would require iteration to reach an optimal
solution. However, it is not clear how PNs perform the gradient decent method and the iteration mechanism.
The major problem, however, is that the solution to eq. (10) might be negative and the PN output rate (lambda)
has to be positive!

- The B matrix is proposed to replace the A matrix so that the role of LNs can be accommodated into the solution.
However, it is not clear why the solution to Eq. (12) (with the B matrix) is the same as the solution to (10) (with the
A matrix). It needs to be shown that the lambda computed by Eq. (12) converges to the one computed by Eq. (10).
Also, the B matrix is square, implicitly indicating that the numbers of the PNs and LNs is the same. However, this is
not true since there are more PNs than LNs. What would be the performance of the reduced dual circuit if the B matrix
is not square but rectangular, and the ratio of its two dimensions follows the ratio of the numbers of LNs and PNs?

- More on the B matrix. The B matrix is not fixed, and needs to be learned every time a new input is given. As shown
in Fig 3, the proposed method can not always learn the B matrix faithfully. What is the performance of the reduced dual
circuit when the B matrix is poorly learned? Furthermore, B needs to be learned before being used for readout by KCs.
Although, the authors depicted a procedure to deal with this issue of combing learning and readout, there is neither a
justification nor simulations of such a procedure, and it is hard to evaluate its correctness.

Finally, the authors suggest that the connectivity between the PNs and the KCs is, at least structurally, related to the A
matrix. Now, in the recently published article: Caron, S.J.C, Ruta, V., Abbott, L.F. and Axel, R. (2013) Random Convergence
of Afferent Olfactory Inputs in the Drosophila Mushroom Body. Nature 497:113-117, it is argued that the same connectivity
is random. Is there a reason to believe that this does not also apply to the locust?

Originality:
Attempting to bridge an optimization method to the neural circuit of insect's olfactory system.

Clarity:
The paper seems at times to be prepared in a rush. The main message is not clear at the first glance.

Quality:
The authors need to demonstrate that the mapping between the sparse optimization solution and neural circuit makes sense,
and the proposed methods indeed solve the decoding problem.

Notes:
line 43: continuosly -> continuously
line 155: multpliers -> multipliers
line 233: inhibtory -> inhibitory
line 234: odor-evoekd -> odor-evoked
line 259: hte -> the
line 314: intepreted -> interpreted
line 348: perfomance -> performance, in correct -> incorrect
line 351: (B) -> (C)
line 352: matrics -> matrices
line 398: out performs -> outperforms
line 418: alotted -> allotted
Summary: The paper falls short both from a mathematical point of view and from the questionable linking of the math with the biology.

Submitted by Assigned_Reviewer_6

NOTE: Due to the range in reviews, we were asked to discuss this paper. As a result of that discussion, we'd like the authors to address the three following points:

1) How are the firing rates (or some other physiological measure of activity) mapped to the variables y and in particular lambda. Lambda - in the current formulation - has no bounds. How does this affect the interpretation.

2) There is a need for "negative reciprocal" connectivity between the PNs and LNs (this comes out of the use of B and B_transpose) in Equation 12. Similarly there is a need for the same symmetry between the measurements of the sparse vector x and the projections up to the KCs. Is there evidence of this symmetry in the locus anatomy or physiology? Can the model work without it?

3) So far the main evidence that this model describes the olfactory circuit in locust is that the odor input is sparse in terms of molecules per input, somewhere along the way the sparse signal is measured by the A matrix such that it is projected into a lower-dimensional, denser representation. (It should be clarified where this happens... we believe it happens on the level of the ORs such that each OR responds to several molecules and each molecule binds with several ORs). Then there is some processing in this lower-dimension via the B matrix (PNs and LNs) which is then projected back up into a higher-dimension through A to the KCs. We feel that while there is some structure of the biological circuit in the model, this paper needs to provide more evidence. Is there evidence of gradient descent dynamics? Is there evidence in the plasticity/processing between LNs and PNs that can be mapped to the form of ICA presented here?

THE TEXT BELOW IS MY ORIGINAL REVIEW:

Firstly, I love seeing a theoretical paper which attempts to model a very particular circuit. Typically, we see attempts to create general neural networks or to formally characterize systems which are not well characterized from a biological/experimental point of view. Here we have the opposite, which seems like a more fruitful approach, that is if one's goal is to understand nervous systems.

In this paper, the authors derive a mathematical model of processing in the locust olfactory system. The input is a bit vector representing an odor where each entry encodes the presence or absence of particular molecules. Because many ORNs each with the same OR converge on a single glomeruli, the input can be taken as noise-free. In the low-dimensional glomerular layer, the mixed signals from the ORNs interact with each other via local interneurons and PNs. Finally, the PNs then project up into the higher-dimensional space of KCs where each KC codes for a particular molecule in the input.

The authors first describe the problem of recovering the sparse, high-dimensional input via the KCs as a MAP optimization, but then reformulate the problem in the compressed sensing context. Next, the dual of the compressed sensing problem is presented as a model since it can more easily be interpreted in terms of the locust nervous system. This model is then changed again to further take knowledge about the biological circuit into account by ensuring that any dynamics happen on the M-dimensional signal. This requires the circuit to learn an M-dimensional subspace of the full N-dimensional space to operate in which is implemented using ICA.

This paper is highly original in the way it brings together the compressed sensing problem, ICA, and a particular neural circuit. Also, the presentation of the model is clear and easy to follow. However, I would like a clearer explanation of the biological circuit and how it maps back to the model. Is it that each OR can bind with several molecules such that the glomeruli each represent a mixture of molecules? Then the PNs readout the glomeruli and also take part in the dynamics which ultimately allows the readout by the KCs which sample a mixture of PNs? The basic question is where exactly does the mixing happen. After reading this paper and reviewing references, I'm still not completely certain. I think where I get lost is whether each OR binds with several or only one molecule. It would be helpful to really nail down how the model and biology map to each other since this paper's strongest aspect is that it models a particular circuit. Maybe to make room you can condense the sections where you present models that are ultimately not used.

As it stands, the current model is very interesting and I believe that with some clarifiations it is motivated by the biology. However, the real key to making this a significant paper would be to show that the dynamics of the model (either the gradient descent dynamics or something about the ICA dynamics) match the dynamics of the true circuit. So far you've shown that this model can perform sparse regression/compressed sensing and that it has some structural similarities to the biological circuit. I'd like to see a figure that compares a simulation of this model with the true circuit even if it's only that individual neuron's responses match physiological data. Or maybe there is evidence of an ICA implementation in the locust olfactory system? Do they dynamics of ICA seem comparable to any behavioral measurements from locust? Or maybe the ability to discriminate odors wouldl show that performance was more like the reduced dual than the full dual circuit? I would love to see something like this addressed in a revised version of this paper or in a future paper.
Summary: This paper presents a novel model for a particular circuit - the locust olfactory circuit - which combines ideas from compressed sensing and ICA. While stronger evidence that this model in fact captures the physiology of the biological circuit would greatly increase the significance of the work, it would still be of interest to the NIPS community as it stands.

Submitted by Assigned_Reviewer_21

Summary:

The authors study the olfaction circuit in the locust. They propose that KC neurons reconstruct estimates of odor vectors via dynamics in the antennal lobe. Specifically, they propose that LN-PN connections are updated via ICA, while the PN dynamics are determined by gradient descent of the dual of a compressed-sensing optimization problem.

They show performance of this "reduced dual" circuit offers reasonable performance compared to a simple feedforward circuit.

Quality:

The reduced dual circuit is an intriguing hypothesis. If true this would be an exceptional paper. The mathematical development flows from 1) basic MAP inference for identifying odors to 2) low-dimensional dynamics of the problem's dual to 3) the reduced dual circuit and finally to 4) learning LN representations by ICA. This step-by-step development is elegant and is thus very appealing.

However, the plausibility of this hypothesis does not seem sufficiently explored. The paper contains no compelling comparisons to biological data. PN neurons are known to exhibit transient dynamics in response to odors. Do the dynamics of eqn 12 yield sensible dynamical trajectories? Can anything be said about physiological/behavioral adaptations of locusts to new odor environments -- and whether such experimental observations comply with expectations of eqn 16? If no such data are available, it would be useful to discuss which expectations of the reduced dual circuit can be tested with existing data and which are currently hypothetical. Overall, these concerns are underdeveloped in the paper.

What is the motivation to focus specifically on the locust? It seems that the locust is only constraining the choice of M and N.

The paper has one unfinished figure (line 294)

Clarity:

In section 4 some notation is confusing. x is overloaded to represent both the odor vector and the KC estimate of the odor. I believe this is motivated by developing the general mathematical result first and adding biological mappings later, but it was slightly confusing. Another example: A represents both the ORN->glom map and the PN->KC map.

Otherwise the paper is clear and well-written.

Originality:

To my knowledge this is an especially original paper.

Significance:

The significance of this study seems to rely on experimental verification. Conditioned on that, I think it has potential to be a high-impact paper.
Summary: Simple biological constraints motivate an alternative formulation for a standard optimization problem (compressed sensing) in locust olfaction. The development of these ideas is elegant, but the paper would be strengthened by verifying expectations of the model with current experimental data.

Submitted by Assigned_Reviewer_37

The paper proposed an inferential and read-out circuit model for decoding odor in locusts. In this circuit, the PN and LN neurons in the antennal lobe engage in inferential computation and the Kanyon cells in the mushroom body perform read-out of the distinct odors. Biological data were taken into consideration in the development of a mathematical model of the circuit's computation, leading to the formulation of the reduced dual circuit, which uses a recurrent circuit between PN cells (lambda), glomerulus activity (y) and LN neurons, which implement the B matrix, to infer and represent the odor in distributed form. The projection from PN activity to KC allows the odor to be read out in sparse representation, with one KC represents roughly one or a few odors. Although some of these ideas about the olfactory circuit have been suggested earlier by G. Laurent and colleagues. This might be the first time the entire computational circuit is put together and mapped into a plausible functional and mathematical form. The insight of using PN and LN cells in recurrent circuit to perform inference is forced by a serious consideration of the biological facts. This model thus provides significant insight into our understanding of olfactory computation, with a fairly plausible and rigorous mathematical framework. The hypothetical connections from LN to PN and their relationship to the projection from PN to KC also provides concrete testable predictions of the model. While parts of this dense paper might need clarification, the work is original and important.
Summary: A novel mathematical model of the olfactory computation circuit that takes both computational functions as well as biological facts into account. It provides deep insight into the computations of the olfactory circuits.
Author Feedback
Author rebuttal: We thank the reviewers for their detailed feedback which we found very useful for improving the clarity and the potential impact of our submission. Below we address the reviewers' comments. We've answered questions specific to a single reviewer individually. Several reviewers asked for clarification in the mapping to the biological circuit, which we will provide in our revised submission. Other questions common to several reviewers are addressed in our response to the Meta Review.

* Reviewer_21
1. "What is the motivation to focus specifically on the locust?"

We selected the locust for concreteness. We expect the same principles to apply to other insects since the relative disparity between M and N is the same.

* Reviewer_4
1. "not clear how PNs perform gradient descent"

The PN dynamics themselves (eq. 10/12) perform gradient descent, since the rate of change of the PN activity vector is proportional to the gradient of the dual Lagrangian. When the dynamics reach a fixed point, the solution has been found. We will clarify the mapping of gradient descent to dynamics in our submission. A mapping of these dynamics in the model to experimentally measured time courses of PN activities is a logical extension of our work - we will expand on this in section 8.1 where we already make some relevant suggestions (see also point 5 and meta review point 3 below).

2. convergence of lambda

The solutions of eq. 10 and 12 are similar but not identical, otherwise the performance of the full and dual systems would be identical, since readout by KCs is performed identically for both. We will further describe the relationship between the two sets of solutions in our updated submission.

3. square B matrix and the numbers of the PNs and LNs

We assumed a square B matrix mainly for mathematical convenience. The performance of the reduced dual circuit using the actual ratio of LNs to PNs would be the same, since all that is required is that that B contain the columns of A that were combined to produce the observations y. Since the sparseness assumption makes the number of columns that were combined low (<10), this requirement can still be met. Nevertheless, this is an important point and we will discuss it in our updated submission.

4. performance of the reduced dual circuit with poorly learned B matrix

ICA as used in our paper can learn the B matrix up to permutation and sign ambiguity. The latter ambiguity is caused by our relaxation of x from binary to real valued when performing ICA. We've performed further work since our submission which allows x to retain its binary nature during ICA, resolving this ambiguity and allowing B to be learned perfectly. We will include this updated implementation of ICA in our final submission.

5. justification for our suggestion for combining learning and readout

Our suggested procedure exploits changes in the electrical properties of LNs during a cycle that would facilitate ICA updates and readout. The viability of this scheme would require a mapping of this work to spiking neurons, which is an important goal for future work (see also point 1 above, and meta review point 3 below)

6. relation to Caron et al. 2013 demonstrating that PN to KC connectivity is random

This is a very interesting point that we also considered before submission. In fact, the same analysis used by Caron et al applied to a binarized version of our connectivity matrix would also yield random connectivity, as we will show in the supplemental information for our revised submission, so there is no inconsistency between the two lines of work.

* Meta Reviews
1. mapping of firing rates to variables

In practice, neither the glomerular activity y nor the PN activity lambda grows very large due to the effects of averaging random tunings, and the increased inhibition by LNs, respectively. The fact that these rates can be both positive and negative while the physiological rates are strictly non-negative can be interpreted as deviations from a baseline rate, as is common in theoretical models (see e.g. Dayan and Abbott 2001). We will include a discussion of this point.

2. "negative reciprocal" connectivity between the PNs and LNs

Only structural information is available about the PN to LN connectivity, specifically that the average PN-to-LN and LN-to-PN connectivity probability is 50-70% (Roni Jortner, personal communication). This is consistent with negative reciprocal connectivity, but experiments must be performed to measure the strength of the connectivity directly. There are also no data available on the tunings of receptors in the locust, and the symmetry between these and the projections to the KCs is a key prediction of our paper. Future work will determine whether these symmetry conditions can be relaxed. We will make these predictions crisper in our dicussion.

3. evidence for gradient descent dynamics / additional biological evidence for the model

The projection to a lower dimension is indeed carried out at the ORs via the affinity matrix, as the reviewers describe. Figure 2D shows temporal patterning in the evoked firing rates of some PNs, consistent with, though not to the same degree as in real PNs. The current evidence for gradient descent, ICA, etc. is very limited because experiments have not explicitly looked for these computations yet, and a contribution of our work is to make specific, experimentally testable predictions regarding these. Supporting evidence for gradient descent is that upon extended odor presentation the PN activity vector settles to a fixed point (Mazur, Laurent 2005). There is also evidence for significant plasticity in the AL (Stopfer, Laurent 1999), as required by our proposal, but whether this is consistent with what we predict for ICA has yet to be tested. We will highlight the current biological evidence and the future experiments needed to test the predictions of our model in the discussion.